# Tolerogenic Dendritic Cells Induce Apoptosis-Independent T Cell Hyporesponsiveness of SARS-CoV-2-Specific T Cells in an Antigen-Specific Manner

**DOI:** 10.3390/ijms232315201

**Published:** 2022-12-02

**Authors:** Mats Van Delen, Ibo Janssens, Amber Dams, Laurence Roosens, Benson Ogunjimi, Zwi N. Berneman, Judith Derdelinckx, Nathalie Cools

**Affiliations:** 1Laboratory of Experimental Hematology, Vaccine & Infectious Disease Institute (VAXINFECTIO), Faculty of Medicine and Health Sciences, University of Antwerp, 2610 Antwerp, Belgium; 2Laboratory of Clinical Biology, Antwerp University Hospital, 2650 Edegem, Belgium; 3Centre for Health Economics Research & Modeling Infectious Diseases (CHERMID), VAXINFECTIO, University of Antwerp, 2610 Antwerp, Belgium; 4Department of Paediatrics, Antwerp University Hospital, 2650 Edegem, Belgium; 5Antwerp Center for Translational Immunology and Virology (ACTIV), VAXINFECTIO, University of Antwerp, 2610 Antwerp, Belgium; 6Antwerp Unit for Data Analysis and Computation in Immunology and Sequencing (AUDACIS), University of Antwerp, 2020 Antwerp, Belgium; 7Center for Cell Therapy and Regenerative Medicine, Antwerp University Hospital, 2650 Edegem, Belgium; 8Department of Neurology, Antwerp University Hospital, 2650 Edegem, Belgium

**Keywords:** tolerogenic dendritic cells, tolerance, severe COVID-19, antigen specificity, T cell apoptosis

## Abstract

Although the global pandemic caused by the novel severe acute respiratory syndrome coronavirus 2 (SARS-CoV-2) is still ongoing, there are currently no specific and highly efficient drugs for COVID-19 available, particularly in severe cases. Recent findings demonstrate that severe COVID-19 disease that requires hospitalization is associated with the hyperactivation of CD4^+^ and CD8^+^ T cell subsets. In this study, we aimed to counteract this high inflammatory state by inducing T-cell hyporesponsiveness in a SARS-CoV-2-specific manner using tolerogenic dendritic cells (tolDC). In vitro-activated SARS-CoV-2-specific T cells were isolated and stimulated with SARS-CoV-2 peptide-loaded monocyte-derived tolDC or with SARS-CoV-2 peptide-loaded conventional (conv) DC. We demonstrate a significant decrease in the number of interferon (IFN)-γ spot-forming cells when SARS-CoV-2-specific T cells were stimulated with tolDC as compared to stimulation with convDC. Importantly, this IFN-γ downmodulation in SARS-CoV-2-specific T cells was antigen-specific, since T cells retain their capacity to respond to an unrelated antigen and are not mediated by T cell deletion. Altogether, we have demonstrated that SARS-CoV-2 peptide-pulsed tolDC induces SARS-CoV-2-specific T cell hyporesponsiveness in an antigen-specific manner as compared to stimulation with SARS-CoV-2-specific convDC. These observations underline the clinical potential of tolDC to correct the immunological imbalance in the critically ill.

## 1. Introduction

Since the first confirmed cases of Severe Acute Respiratory Syndrome Coronavirus-2 (SARS-CoV-2) infections in December 2019 [1], there have 629,978,289 confirmed cases worldwide, with 6,582,023 confirmed deaths (situation 9 November 2022) [2]. Accordingly, Corona Virus Disease-2019 (COVID-19) has been declared a global pandemic as of 11 March 2020 [1]. Interestingly, the clinical image of COVID-19 displays a heterogeneous range from mild symptoms in most of the infected patients to a moderate or severe disease progression requiring hospitalization or admission to an intensive care unit (ICU) in 19% of infected patients [3,4]. Several studies have reported a decrease in peripheral T cell counts during the acute phase of infection [5,6]. More detailed phenotypical analysis revealed that this depletion was accompanied by a higher activation and reactivity of remaining IFN-γ-producing T cells proportionate with disease severity [7,8]. Patients with severe COVID-19 often show a profound dysregulation of the immune response [9], as evidenced by a shift in the cytokine profile of severe COVID-19 patients towards an increase in proinflammatory cytokines such as interleukin (IL)-6, IL-8, IL-17 and macrophage inflammatory protein (MIP-1α) [10]. Ghazavi et al. [11], on the other hand, only found a significant upregulation of Transforming Growth Factor (TGF)-β.

Indeed, an extensive fraction of patients with a severe disease course display an over-activation of the immune system against (parts of) the SARS-CoV-2 virus. Using single-cell transcriptomics, a study by Kalfaoglu et al. [12] demonstrated that CD4^+^ T cells from severe COVID-19 patients differentially expressed activation-induced genes, such as a higher expression of *JUN* and *MKI67* as compared to CD4^+^ T cells from moderate COVID-19 patients, indicative of a substantially high immune activation status in severe COVID-19 patients. In addition, severe COVID-19 patients showed the transcriptional suppression of the expression of Forkhead Box P3 (FOXP3) and an impaired differentiation of Th17 cells compared to moderate COVID-19 patients [12]. A prospective cohort study by Gadotti et al. [13] showed a positive correlation between sustained high interferon-γ (IFN-γ) levels over time and poorer prognosis in severe COVID-19 patients. In addition, Kang et al. [14] compared the phenotype and function of T cells isolated from the peripheral blood mononuclear cells (PBMC) of severe COVID-19 patients at the first vs. the third week of the disease course. They found a high expression of the cytotoxicity markers Perforin and Granzyme B and an increase in proliferation and activation markers in CD4^+^ and CD8^+^ T cells over time, potentially related to a hyperactive state of the cell-mediated immune response [14]. Particularly in severe COVID-19 patients, higher frequencies of SARS-CoV-2-specific IFN-γ^+^CD69^+^CD4^+^ T cells, with a high expression of Cytotoxic T-Lymphocyte-associated Protein 4 (CTLA-4), were observed as compared to uninfected and convalescent patients [15]. According to a study by Weiskopf et al. [16] in severe COVID-19 patients, the majority of CD4^+^CD69^+^CD137^+^ SARS-CoV-2-specific T cells were restricted to a central memory subtype and showed a high secretion of the effector cytokines IFN-γ, IL-2 and tumor necrosis factor (TNF)-α. Similarly, in long COVID-19, higher frequencies of IFN-γ and TNF-α secreting SARS-CoV-2-specific T cells were observed as compared to patients with resolved COVID-19 [17]. In the absence of an effective SARS-CoV-2-specific therapy, treatment options are predominantly symptomatic-oriented, including infection prevention and supportive care [18]. The use of non-antigen-specific immunosuppressive drugs for the treatment of severe COVID-19 has been investigated, with several immunomodulatory drugs being proposed [19]. Dexamethasone has been shown to decrease mortality in severe COVID-19 patients who require respiratory support [20]. However, the use of an antigen-specific suppression of the harmful hyperinflammatory T cell response could be an interesting avenue to pursue. Altogether, these data indicate a remarkable state of hyperactivation in the T cell compartment and a drastic alteration of the immune response in patients with a severe COVID-19 progression, underlining the urgent need for immunomodulating therapies capable of specifically downmodulating the hyperactivated T cell response.

Dendritic cells (DCs) are often considered key players in the balance between immunity and tolerance [21]. These professional antigen-presenting cells can induce a strong antigen-specific immune response to foreign antigens, as well as tolerance to self-antigens. As such, DCs provide prospects for immune activation as well as for immunosuppressive therapies [22]. Currently, the use of tolerogenic dendritic cells (tolDCs) for the treatment of several inflammatory autoimmune diseases, such as multiple sclerosis (MS) and rheumatoid arthritis (RA), is being investigated [23,24]. 

Various strategies have been used to generate tolDC in vitro, including (1) pharmaceuticals (e.g., vitamin D_3_ (vitD_3_) [25] or rapamycin [26]), (2) biologicals (e.g., IL-10 [27] or TGF-β [28]) and (3) genetic engineering (using antisense oligonucleotides against CD40, CD80 and CD86 [29]). Although these different methods have the same purpose, the resulting product has common characteristics indicative of tolerogenic properties with method-specific differences [30]. The tolerogenic phenotype is one of the most recognized characteristics of tolDC. It is generally accepted that tolDC display a stable semi- or immature phenotype that is resistant to maturation [31]. Among others, there is a reduced expression of co-stimulatory molecules such as CD80 and CD86 and a reduced expression of the DC maturation marker CD83 [32]. In addition, the production of cytokines shifts from an inflammatory profile to an immunosuppressive profile (e.g., the upregulation of IL-10, TGF-β) [30]. Functionally, tolDC suppress effector T cells through different mechanisms, such as the inhibition of T cells by the induction of anergy [33], T cell deletion by apoptosis [30], the induction of regulatory T cells [34] or regulatory B cells [35] or the promotion of a tolerogenic environment by cytokines or metabolites [36], depending on the method by which they were generated [37]. In the current study, tolDC were generated by treatment with vitD_3_, as previously described by Lee et al. [25].

In the present study, we aimed to investigate whether tolDC can induce T cell hyporesponsiveness in SARS-CoV-2-specific T cells in an antigen-specific manner. For this, SARS-CoV-2-specific T cells were isolated from COVID-19-seropositive donors and stimulated in vitro with tolDC. To our knowledge, this study is the first evaluating the use of vitD_3_-treated tolDC to induce T cell hyporesponsiveness of SARS-CoV-2-specific T cells. In doing so, we anticipate providing a proof of concept for the use of tolDC to counteract T cell hyperinflammation in severe COVID-19. Furthermore, as much remains to be understood about the molecular and cellular function of tolDC, we evaluated whether T cell hyporesponsiveness is mediated by tolDC via the initiation of antigen-specific T cell deletion. With this study, we ultimately aim to contribute to the current understanding of the mode of action of tolDC and will provide future prospects of novel therapeutics for severe COVID-19.

## 2. Results

### 2.1. VitD_3_-Treated tolDC Display a Semimature Phenotype and Induce T-Cell Hyporesponsiveness in an Allo-MLR

DC were differentiated from monocytes in the presence of GM-CSF and IL-4 (convDC). Additionally, vitD_3_ was added for the generation of tolDC. At day 6 of the cell culture, tolDC and convDC were harvested with a viability of, respectively, 65% (interquartile range (IQR): 60–70%) and 84% (IQR: 83–87%). In the samples, a high proportion of CD209 (DC-SIGN)-expressing cells (convDC: 97%, IQR: 95–98% and tolDC: 96%, IQR: 94–97%) and HLA-DR-expressing cells (convDC: 100%, IQR: 100–100% and tolDC: 98%, IQR: 95–99%) was observed, confirming DC identity (Figure 1A). Interestingly, a significantly higher proportion of CD14-expressing tolDC (1.3%, IQR: 1–6%) was found as compared to CD14-expressing convDC (0.3%, IQR: 0.1–0.4%; *p* = 0.03). Most CD14^+^ cells co-expressed CD209, and only negligible numbers of contaminating CD209^−^CD14^+^ monocytes were present in cell cultures (Figure 1A).

Next, the expression levels of the maturation-associated markers CD80, CD83 and CD86 were evaluated on tolDC and convDC (Figure 1C). The expression of CD80 was significantly decreased in tolDC as compared to convDC (*p* = 0.03), with an average decrease of 58% ± 22%. Additionally, the expression of CD83 (*p* = 0.03) and CD86 (*p* = 0.03) was significantly decreased in tolDC as compared to convDC, with an average decrease of 70% ± 10% and 66% ± 11%, respectively (Figure 1B, Appendix A). Although we could not find a difference in the proportion of HLA-DR expressing DC, HLA-DR expression levels were significantly decreased (*p* = 0.03) in vitD_3_-treated tolDC as compared to convDC, with an average decrease of 73% ± 9.7% (Figure 1B, Appendix A).

Finally, the T cell stimulatory capacity of DC was assessed by means of allo-MLR, in which PBL were stimulated with allogeneic DC in a 10:1 ratio. IFN-γ secretion by T cells was analyzed using IFN-γ ELISA and was used as a measure for T cell stimulation. As shown in Figure 1D, significantly lower levels of IFN-γ were secreted by PBL stimulated with allogeneic tolDC as compared to PBL stimulated with allogeneic convDC (*p* = 0.03). The T cell stimulatory capacity of tolDC was reduced by 69% ± 20%, on average, as compared to convDC.

In summary, our observations demonstrate the successful generation of vitD_3_-treated tolDC with a semimature phenotype, as evidenced by reduced expression levels of the DC activation markers CD80, CD83 and CD86 and by the reduced T cell stimulatory capacity.

### 2.2. TolDC Induce Antigen-Specific T Cell Hyporesponsiveness in SARS-CoV-2-Specific T Cells

To evaluate whether the function of SARS-CoV-2-specific T cells can be attenuated by tolDC, SARS-CoV-2-specific T cells were first isolated. For this, PBL from COVID-19 seropositive donors were stimulated with a SARS-CoV-2-derived peptide pool for 6 days and subsequently selected based on the expression of the cellular activation markers CD71 and CD98 (Figure 2A) [38]. A 94% (IQR: 92–96%) pure population of CD71^+^CD98^+^ double-positive lymphocytes was obtained following flow cytometric sorting (Figure 2B), demonstrating an efficient isolation of SARS-CoV-2-specific T cells. Next, SARS-CoV-2-specific T cells were stimulated with tolDC or with convDC, in the presence of SARS-CoV-2 peptides. After 6 days, cultured cells were harvested and rechallenged with SARS-CoV-2 peptides in an IFN-γ ELISpot assay (Figure 2C). A significant increase in SARS-CoV-2-specific IFN-γ-producing cells was observed when SARS-CoV-2-specific T cells were stimulated with convDC in the presence of SARS-CoV-2 peptides (512.5 spot forming units (SFU), IQR: 245.6–706.9, *p* = 0.0125) as compared to SARS-CoV-2-specific T cells stimulated with SARS-CoV-2 peptides alone (97.5 SFU, IQR: 68.4–154.7). Interestingly, when SARS-CoV-2-specific T cells were stimulated with vitD_3_-treated tolDC in the presence of SARS-CoV-2 peptides (137.5 SFU, IQR: 50.0–169.9), a significant decrease in the amount of IFN-γ SFU was observed as compared to stimulation with convDC (*p* = 0.0148, Figure 2D), whereas no significant difference (*p* > 0.9999) was found in comparison with SARS-CoV-2-specific T cells alone (97.5 SFU, IQR: 68.4–154.7, Figure 2D).

Additionally, to investigate whether the induction of SARS-CoV-2-specific T cell hyporesponsiveness by vitD_3_-treated tolDC was antigen-specific, PBL stimulated with vitD_3_-treated tolDC in the presence of SARS-CoV-2 peptides were rechallenged with SARS-CoV-2 peptides or with an irrelevant peptide pool, namely, cytomegalovirus (CMV) pp65 (Figure 2E). Similarly, as with SARS-CoV-2-specific T cells, the stimulation of PBL with vitD_3_-treated tolDC in the presence of SARS-CoV-2 peptides results in a significantly lower number of SFU (187.3 SFU, IQR: 110.2–234.7), as compared to the stimulation of PBL with convDC (445.7 SFU, IQR: 357.8–482.5; *p* = 0.0163), when cells were rechallenged with SARS-CoV-2 peptides. Remarkably, when PBL stimulated with vitD_3_-treated tolDC in the presence of SARS-CoV-2 peptides were rechallenged with CMV pp65 peptides, PBL could mount an adequate amount of IFN-γ SFU (418.7 SFU, IQR: 357.3–493.7). In fact, no significant difference (*p* > 0.9999) was observed in the CMV-specific IFN-γ response when PBL were stimulated with tolDC, as compared to PBL being stimulated with convDC (432.7 SFU, IQR: 387.2–524.2), when cells were rechallenged with CMV pp65 peptides. Altogether, these findings underscore that the T cell hyporesponsiveness mediated by vitD_3_-treated tolDC is antigen-specific.

### 2.3. SARS-CoV-2-Specific T Cell Hyporesponsiveness Induced by tolDC Is Not Mediated by the Apoptosis of CD4^+^ T Cells

To elucidate the mechanism by which tolDC exert their immunomodulatory function, we longitudinally evaluated the induction of T cell deletion during the DC-T cell coculture. In brief, SARS-CoV-2-specific T cells were stimulated with tolDC or with convDC in the presence of SARS-CoV-2 peptides, during which CD4^+^ T cell apoptosis was assessed at 24, 72 and 120 h of co-culture (Figure 3A).

While a statistically significant difference was observed in the percentage of viable (*p* = 0.0465) and late apoptotic SARS-CoV-2 T cells (*p* = 0.0015) in each condition over time, no statistically significant differences were observed over time in the percentage of viable, early apoptotic and late apoptotic SARS-CoV-2-specific CD4^+^ T cells stimulated with tolDC as compared to convDC (Figure 3B, Appendix A). Indeed, the percentage of viable T cells after 24 h did not significantly differ in cocultures with tolDC (77.1%, IQR: 70.7–83.5) when compared to cocultures with convDC (79.4%, IQR: 70.8–87.3), as was also seen after 72 h (tolDC 78.5%, IQR: 67.3–87.6, vs. convDC 86.0%, IQR: 76.6–91.5) and after 120 h (tolDC 73.4%, IQR: 59.5–80.3, vs. convDC 80.2%, IQR: 69.3–84.9). Similarly, no differences in early apoptosis were seen after 24 h (tolDC 11.1%, IQR: 6.1–18.6 vs. convDC 10.3%, IQR: 4.1–18.7), after 72 h (tolDC 8.2%, IQR: 4.3–11.0 vs. convDC 5.6%, IQR: 3.8–9.1) or after 120 h (tolDC 7.8%, IQR: 5.3–18.8 vs. convDC 5.5%, IQR: 3.6–8.6). Lastly, the proportion of T cells that are considered late apoptotic were not significantly different between those cocultured with tolDC and those cocultured with convDC after 24 h (tolDC 9.8%, IQR: 5.9–13.2 vs. convDC 8.6%, IQR: 6.9–11.4), after 72 h (tolDC 10.5%, IQR: 5.1–22.1 vs. convDC 7.0%, IQR: 5.1–13.0) or after 120 h (tolDC 17.1%, IQR: 11.7–24.9 vs. convDC 13.2%, IQR: 10.5–22.2). Each condition also showed no significant difference at each timepoint as compared to SARS-CoV-2-specific T cells alone (Figure 3B and Appendix A). Likewise, intracellular caspase 3/7 levels showed a statistically significant difference over time (*p* < 0.0001), but no statistical differences were observed (Figure 3C, Appendix A), when measured at each timepoint, between SARS-CoV-2-specific CD4^+^ T cells cocultured with tolDC and those cocultured with convDC as well as SARS-CoV-2-specific T cells alone (Appendix A).

In summary, T cells stimulated with vitD_3_-treated tolDC did not show an increase in CD4^+^ T cell apoptosis as compared to T cells cocultured with convDC based on intracellular caspase 3/7 and the cell death markers Annexin-V and SYTOX AADvanced Dead cell stain.

## 3. Discussion

Although most patients infected with SARS-CoV-2 only display a mild disease course, a considerable patient group develops severe COVID-19, which could result in respiratory distress, multi-organ failure and/or death [39]. A systematic review which included 97 studies reported a case-fatality rate of 46 to 62% in patients admitted to the ICU with severe COVID-19 [40]. Patients often present with elevated levels of cytokines related to an effector and T helper 1 (Th1) response and several organ complications [41]. Here, we have provided proof of concept for the use of vitD3-treated tolDC as a potential cell-based therapy for patients with severe COVID-19. These vitD_3_-treated tolDC were characterized by a semimature phenotype and a low T cell stimulatory capacity. Our observations demonstrate that SARS-CoV-2 peptide-pulsed tolDC are capable of inducing SARS-CoV-2-specific T cell hyporesponsiveness in an antigen-specific manner. Furthermore, our data show that the induction of T cell hyporesponsiveness is not mediated by the induction of CD4^+^ T cell apoptosis.

For this, we have first isolated a pure population of SARS-CoV-2-specific T cells. Indeed, we demonstrate the feasibility of isolating SARS-CoV-2-specific CD4^+^ and CD8^+^ T cells following an activation-induced marker (AIM) assay based on CD71 and CD98 expression. Upon in vitro stimulation with SARS-CoV-2 peptides, both CD71 and CD98 are upregulated during T cell proliferation [38], and activated CD71^+^CD98^+^ T cells were sorted with an average purity of 93.88% ± 2.89%. In doing so, we present an efficient and feasible approach to obtaining a pure antigen-specific lymphocyte population, in agreement with previous findings by others demonstrating that the sorting of the double-positive population allows for the efficient isolation of a viable, antigen-specific T cell population [38]. This technique provides an advantage over other activated T cell isolation techniques such as a tracking-dye based approach, since no toxic dye needs to be used [42], or intracellular cytokine staining, in which cells need to be fixed and permeabilized [43]. Besides the use of these isolated SARS-CoV-2-specific T cells in this study, the possibility of isolating SARS-CoV-2-specific T cells could be of high relevance for further research into the role of SARS-CoV-2-specific T cells in COVID-19 disease progression—for instance, enabling gene-expression profiling studies elucidating potential differences between patients with mild COVID-19 and severe COVID-19. Additionally, the technique is easily amendable for isolating other antigen-specific T cell populations for downstream analysis.

Several characteristics and functions of tolDC have been demonstrated to contribute to the tolerogenic function of these cells. As previously shown by others [44,45] and described in this study, tolDC are characterized by a semimature immunophenotype shown by the significantly lower expression of the maturation markers CD80, CD83 and CD86 as compared to convDC. VitD_3_-treated tolDC are highly positive for the expression of CD209 and HLA-DR. However, the expression level of HLA-DR in vitD_3_-treated tolDC was significantly decreased as compared to convDC. This observation is in line with previous research that indicates low expression levels of MHC molecules resulting in low levels of antigen presentation by tolDC [32]. Interestingly, tolDC retained some CD14 expression, which is associated with immunosuppressive functions [46]. Secondly, tolDC showed an impaired T cell stimulatory capacity, as evidenced by a decrease in the pro-inflammatory IFN-γ response following stimulation with tolDC as compared to T cells stimulated with SARS-CoV-2-antigen-loaded convDC. In fact, no difference in the amount of IFN-γ secreted by T cells stimulated with tolDC and T cells challenged by antigen in the absence of antigen-presenting cells was observed, further demonstrating the low T cell stimulatory capacity of tolDC. Importantly, the induced T cell hyporesponsiveness by tolDC was SARS-CoV-2 antigen-specific, as demonstrated by the observation that T cells that were unresponsive towards SARS-CoV-2 following stimulation with tolDC maintained their ability to respond towards an irrelevant peptide. This is of particular interest in immune-mediated diseases with a clear TCR signature suggestive of a superantigen such as Multisystem Inflammatory Syndrome-Children (MIS-C) or Pediatric Inflammatory Multisystemic Syndrome [47,48], a systemic hyperinflammatory disorder often seen in children following SARS-CoV-2 infection [49].

To date, the exact mechanism by which tolDC induce T cell hyporesponsiveness remains unclear [37]. The decrease in the expression levels of HLA-DR by tolDC as compared to convDC is indicative of decreased antigen presentation, which can also contribute to the induction of T cell hyporesponsiveness [50]. However, several studies have shown that DCs can induce a strong and effective T cell response even when low levels of antigen were present [51,52]. Indeed, our results show that vitD_3_-treated tolDC were still able to stimulate a strong T cell response when rechallenged with an irrelevant peptide. It is more likely that other mechanisms are involved in the induction of T cell hyporesponsiveness by vitD_3_-treated tolDC. Previously, we demonstrated that tolDC rendered T cells in a robust hyporesponsive state, thereby excluding the tolDC-mediated induction of T cell anergy, in agreement with others [53]. Moreover, we and others [53,54] were not able to demonstrate the induction or expansion of regulatory T cells (Treg) by vitD_3_-treated tolDC [25,55]. Therefore, in the current study, we hypothesized that vitD_3_-treated tolDC induce T cell hyporesponsiveness via the induction of T cell deletion. Among other factors involved in the apoptotic process, caspase-3 and -7 are considered important effector caspases [56]. Simultaneous staining with 7-amino-actinomycin D (7-AAD), a DNA intercalator, and Annexin-V, which is able to bind to phosphatidyl serine translocated to the outer layer of the cell membrane during apoptosis, provides an efficient method to follow-up cell apoptosis [57]. However, our results indicate that tolDC do not induce substantially more T cell death compared to convDC, although T cell viability decreased over time in both conditions. In addition, similar intracellular expression levels of caspase-3 and caspase-7 [56] were found between SARS-CoV-2 T cells stimulated with tolDC and SARS-CoV-2 T cells stimulated with convDC. In our hands, we did not observe any differences in the induction of T cell deletion by vitD_3_-treated tolDC, albeit we only tested CD4^+^ T cell deletion since the used peptide pool was predominantly MHC class II-oriented and thus only activated small numbers of CD8^+^ T cells. By using a pure antigen-specific T cell population, we were able to surpass hurdles in assessing antigen-specific T cell deletion with the current techniques [58]. Hence, we developed an assay capable of analyzing the induction of T cell apoptosis efficiently.

Since we did not show the involvement of T cell deletion in the mechanism of vitD_3_-treated tolDC, the exact mode of action remains elusive. A recent study has shown that the secretion of a high amount of lactate by tolDC can suppress T cell proliferation [59], as such metabolic reprogramming of T cells remains of interest. In addition, the role of extracellular vesicles in the induction of T cell hyporesponsiveness by tolDC has recently gained interest since it has been established that DC-derived exosomes can reflect the state and function of the cell of origin [60]. In the current study, only the donors’ COVID-19 seropositivity was evaluated, since no information was available regarding the COVID-19 symptomatology and the COVID-19 variant. Whether the severity of the COVID-related symptomatology affects tolDC generation and the induction of hyporesponsiveness remains to be elucidated. However, we have demonstrated that the vitD_3_ treatment of monocyte-derived DC induces a similar tolerogenic DC phenotype in MS patients and healthy controls, despite the hyperactive state of DCs in MS patients [25,61].

Altogether, our results provide in vitro proof of concept of the ability of tolDC to induce SARS-CoV-2-specific T cell hyporesponsiveness. Moreover, this hyporesponsiveness was not mediated by the induction of CD4^+^ T cell apoptosis. Further research into the exact mode of action is needed to fully elucidate the tolerogenic function of tolDC. Recently, the involvement of regulatory B cells and cell-contact-independent mechanisms such as cytokine deprivation and extracellular vesicle-mediated suppression has raised interest [35].

## 4. Materials and Methods

### 4.1. Human COVID-19 Seropositive Blood Samples

Buffy coats from anonymous healthy donors were provided by the Blood Transfusion Center of the Red Cross (Mechelen, Belgium). A total of 3 mL was transferred to an SST™ II Advance Vacutainer^®^ Blood collection tube (BD Diagnostics, Plymouth, UK), and serum was isolated according to the manufacturer’s instructions. COVID-19 serology was tested using the Atellica^®^ IM SARS-CoV-2 IgG (sCOVG) assay (Siemens Healthineers, Beersel, Belgium). In short, SARS-CoV-2-specific IgG antibodies were captured using streptavidin-coated microparticles with biotinylated SARS-CoV-2 recombinant antigens. After the addition of an acridinium-ester-labeled anti-human IgG mouse antibody, the amount of SARS-CoV-2-IgG antibody could be determined by measuring on the Atellica IM Analyzer (Siemens Healthineers). 

### 4.2. Monocyte-Derived Dendritic Cell Culture

Peripheral blood mononuclear cells (PBMC) were isolated from COVID-19-seropositive buffy coats by Ficoll-Paque density gradient centrifugation (GE Healthcare, Diegem, Belgium). Next, CD14^+^ monocytes were isolated using CD14-microbead-based immunomagnetic selection (CD14 reagent, Miltenyi Biotec, Leiden, The Netherlands), according to the manufacturer’s instructions. To generate conventional monocyte-derived DC (convDC), purified CD14^+^ cells were resuspended in Iscove’s Modified Dulbecco’s Medium (IMDM, Life Technologies, Merelbeke, Belgium), supplemented with 2% human AB serum (hAB; Life Technologies), 200 IU/mL granulocyte-macrophage colony-stimulating factor (GM-CSF, Gentaur, Kampenhout, Belgium) and 250 IU/mL interleukin (IL)-4 (Miltenyi Biotec) at day 0 and cultured at a concentration of 1 × 10^6^ cells/mL for 6 days in culture flasks (Greiner Bio-One, Vilvoorde, Belgium). Simultaneously, tolDC were generated under the same conditions, but with the addition of 2 nM vitamin D_3_ (VitD_3_, EcoPharma Supply, Breda, The Netherlands) on day 0 (start of culture) and on day 4. On day 4, both convDC and tolDC were stimulated for 48 h with a cocktail of pro-inflammatory cytokines consisting of 1000 IU/mL IL1-β (Miltenyi Biotec), 1000 IU/mL tumor necrosis factor (TNF)-α (Miltenyi Biotec) and 2.5 μg/mL prostaglandin E_2_ (PGE_2_; Pfizer, Puurs, Belgium). All cells were cultured in a humidified atmosphere with 5% CO_2_ at 37 °C. On day 6, convDC and tolDC were harvested and used in further experiments. Viability was assessed by flow cytometry using propidium iodide (PI, Thermo Fisher Scientific, Merelbeke, Belgium) staining. For analytical flow cytometry, 10^4^ events were acquired using a Cytoflex flow cytometer (Beckman Coulter, Analis, Suarlée, Belgium).

The CD14-negative fraction, i.e., peripheral blood lymphocytes (PBL), was cryopreserved for further use. For this, 250 × 10^6^ CD14^−^ cells were resuspended in fetal bovine serum (FBS; Life Technologies) supplemented with 10% dimethylsulfoxide (DMSO; Sigma-Aldrich, Diegem, Belgium) at a concentration of 50 × 10^6^ cells/mL. All aliquots were frozen using Corning^®^ Coolcell™ LX cell-freezing containers (Corning, Lasne, Belgium) at −80 °C.

### 4.3. Immunophenotyping of DC

The phenotypical analysis of tolDC and convDC was implemented as part of the quality analysis of the cell product. Immunofluorescence staining was performed using the following fluorochrome labeled monoclonal antibodies (mAb): anti-CD209-fluorescein isothiocyanate (FITC, BD Pharmingen, Erembodegem, Belgium), anti-HLA-DR-phycoerythrin (PE, BD Biosciences, Erembodegem, Belgium), anti-CD14-peridinin chlorophyll (PerCP, BD Biosciences), anti-CD80-PE-Cyanine5 (Cy5, BD Pharmingen), anti-CD83-FITC (Life Technologies) and anti-CD86-PE (BD Pharmingen, Appendix A). Isotype-matched control antibodies were used to determine non-specific background staining (Appendix A). For analytical flow cytometry, 10^4^ events were measured based on the light scatter properties of DC by means of forward scatter (FSC) and side scatter (SSC).

The decrease in mean fluorescence intensity (MFI) is calculated as follows:% decrease = (1 − (MFI tolDC/MFI convDC)) × 100

### 4.4. Allogeneic Mixed Lymphocyte Reaction (Allo-MLR)

To assess the allogeneic T cell stimulatory capacity of tolDC and convDC, allogeneic PBL were stimulated with either tolDC or convDC at a 10:1 ratio. Non-stimulated allogeneic PBL served as a negative control. Cocultures were performed in IMDM supplemented with 5% hAB serum in a humidified atmosphere with 5% CO_2_ at 37 °C. After 5 days, the secreted level of interferon-γ (IFN-γ) in the cell culture supernatant was determined using a commercially available enzyme-linked immunosorbent assay (ELISA; PeproTech, London, UK) as a measure for the T cell stimulatory capacity of DC. Each experiment is measured in triplicate. Plates were read by measuring absorbance at 405 nm using a Victor^3^ multilabel plate reader (PerkinElmer, Mechelen, Belgium) and interpolated to the concentration (pg/mL) using MS Office Excel.

### 4.5. In Vitro Activation and Isolation of SARS-CoV-2-Specific T Cell Populations

For the in vitro expansion, 50 × 10^6^ PBL were resuspended in IMDM supplemented with 5% hAB at a concentration of 4 × 10^6^ cells/mL. On day 0, a 2 μg/10^6^ cells SARS-CoV-2 SNMO (S, spike protein; N, nucleoprotein; M, membrane protein; O, open reading frame) defined peptide pool (Mabtech, Nacka Strand, Sweden) was added to the PBL. As a positive control, 10 × 10^6^ PBL were stimulated with 1 μg/mL *Staphylococcal* enterotoxin B (SEB; Sigma-Aldrich), whereas PBL were left unstimulated as a negative control. Cells were cultured in a humidified atmosphere with 5% CO_2_ at 37 °C. After 6 days, cells were harvested and stained with the following fluorochrome-conjugated mAb: anti-CD3-PerCPCy5.5 (Biolegend, Amsterdam, The Netherlands), anti-CD4-Brilliant Violet (BV) 510 (Biolegend), anti-CD8-Pacific Blue (PB, Biolegend), anti CD71-BV786 (Biolegend) and anti-CD98-Brilliant Blue (BB) 515 (Biolegend, Appendix A), and viability was assessed using the LIVE/DEAD fixable Near-IR Dead Cell Stain Kit (ThermoFisher Scientific, Merelbeke, Belgium). Activated SARS-CoV-2-specific CD4^+^ and CD8^+^ T cells were identified, respectively, as CD3^+^CD4^+^CD71^+^CD98^+^ T cells and CD3^+^CD8^+^CD71^+^CD98^+^ T cells [38] and isolated by means of Fluorescence Activated Cell Sorting (FACS). Cells were sorted as lymphocytes on scatter (FSC-A, SSC-A), single cells (FCS-A, FSC-H) and viable cells (Fixable Near-IR Dead Cell Stain), activated CD4^+^ T cells were sorted as CD3^+^, CD4^+^, CD71^+^ and CD98^+^ and activated CD8^+^ T cells were sorted as CD3^+^, CD8^+^, CD71^+^ and CD98^+^ (Figure 2A) using a BD FACSAria II device (BD Biosciences). After sorting, an aliquot of the minimum 1 × 10^4^ of the sorted cells was used to confirm purity (Figure 2B). The number of cells was determined using a hematological cell counter (ABX Micros 60, Horiba, Diegem, Belgium). 

### 4.6. DC-T Cell and DC-PBL Coculture

To investigate the T cell stimulatory capacity of DC, in vitro-generated DC (either tolDC or convDC) and isolated activated SARS-CoV-2-specific T cell fractions were stimulated with in vitro-generated DC (either tolDC or convDC) at a 1:10 ratio for 5 days in IMDM supplemented with 5% hAB in the presence of a 2 μg/10^6^ cells SARS-CoV-2 SNMO defined peptide pool. To determine the antigen specificity, an identical coculture was set up in which PBL were used instead of T cells. To assess whether tolDC induce T cell deletion during the DC-T cell coculture, a sample was taken every 48 h, and immunofluorescence staining was performed using the following fluorochrome-labeled mAb: anti-CD3-PECy7 (Biolegend), anti-CD4-allophycocyanin-H7 (Biolegend) and anti-CD8-PB (biolegend), in combination with the Annexin-V-APC and CellEvent™ Caspase-3/7 Green Flow Cytometry Assay Kit (Figure 3A, Appendix A). The results were acquired on the Novocyte Quanteon™ flow cytometer (Agilent, Machelen, Belgium). For analytical flow cytometry, 10^4^ events were measured based on the light scatter properties of T cells by means of forward scatter (FSC) and side scatter (SSC). In the analysis, viable cells were defined as Annexin-V^−^SYTOX™AADvanced^−^, early apoptotic cells were defined as Annexin-V^+^SYTOX™AADvanced^−^ and late apoptotic cells were defined as Annexin-V^+^SYTOX™AADvanced^+^. On day 5, cells were harvested to analyze antigen-specific T cell activity by means of IFN-γ ELISpot. Cells were harvested and a maximum of 1 × 10^5^ cells/well were seeded in a 96-well MultiScreen-IP Filter Plate (Merck Millipore, Overijse, Belgium) in triplicate in IMDM supplemented with 5% hAB. T-cells were restimulated with a 2 μg/10^6^ cells SARS-CoV-2 SNMO defined peptide pool or a 5 μg/10^6^ cells cytomegalovirus (CMV) pp65 peptide pool (NIH HIV Reagents program). After 22 h, the plates were further processed using the commercially available human IFN-γ ELISpot^BASIC^ kit (Mabtech). Finally, ELISpot plates were counted using the AID iSpot Reader System (AID Diagnostika, Straßberg, Germany).

To assess the CMV reactivity, only CMV-responsive donors were selected for analysis. A donor was a CMV responder if the difference between the number of SFU in the CMV-stimulated condition was greater or equal to 15 spots as compared to the unstimulated condition and a ratio of at least 1.5 between the number of SFU in the CMV-stimulated condition compared to the unstimulated condition was present.

### 4.7. Statistical Analysis

Flow cytometric data were analyzed using FlowJo 10.7.1 Software (Flowjo™, TreeStar Inc, Ashland, OR, USA). Unless stated otherwise, values are given as the median, with IQR in the form of the median (Q25-Q75). Non-parametrical statistical analysis was performed with GraphPad Prism 9.1.2 using a Wilcoxon signed rank test or Kruskal Wallis test followed by a post-hoc Dunn’s multiple comparison test, when applicable. During the coculture, a mixed effect model with the Geisser-Greenhouse correction and matched values stacked in a sub-column was performed. A *p*-value < 0.05 was considered statistically significant.

## 5. Conclusions

TolDC are capable of inducing SARS-CoV-2-specific T cell hyporesponsiveness in vitro in an antigen-specific manner. This hyporesponsiveness was not mediated by the induction of T cell apoptosis. Hence, further research into the exact mode of action is needed to fully elucidate the tolerogenic function of tolDC. Ultimately, this study may provide new insights into future therapeutics for severe COVID-19.

## Figures and Tables

**Figure 1 ijms-23-15201-f001:**
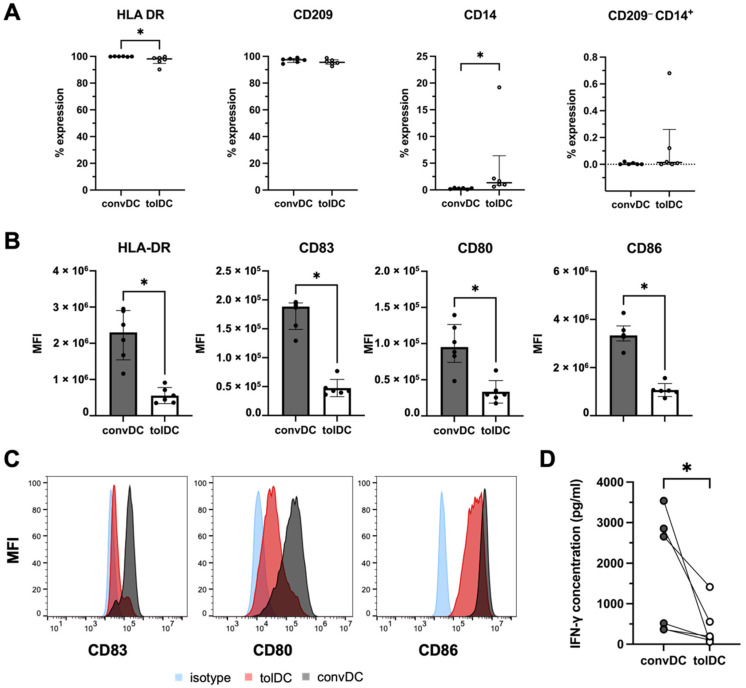
TolDC display a semimature phenotype and reduced T cell stimulatory capacity. (**A**) The percentage of cells expressing the markers HLA-DR, CD209 and CD14 and the percentage of cells that are CD209^neg^CD14^pos^. TolDC are indicated in white dots; convDC are indicated in black dots (n = 6). (**B**) Expression levels of HLA-DR, CD83, CD80 and CD86 shown as MFI values of tolDC (white bars) or convDC (black bars) (n = 6). (**C**) Representative histogram overlay of CD83, CD80 and CD86 expression levels by tolDC (red), convDC (black) and isotype control (blue). (**D**) Concentration in pg/mL of IFN-γ secretion by PBL stimulated with convDC (black dots) and tolDC (white dots) (n = 6). Significance (*p* < 0.05) is indicated by *.

**Figure 2 ijms-23-15201-f002:**
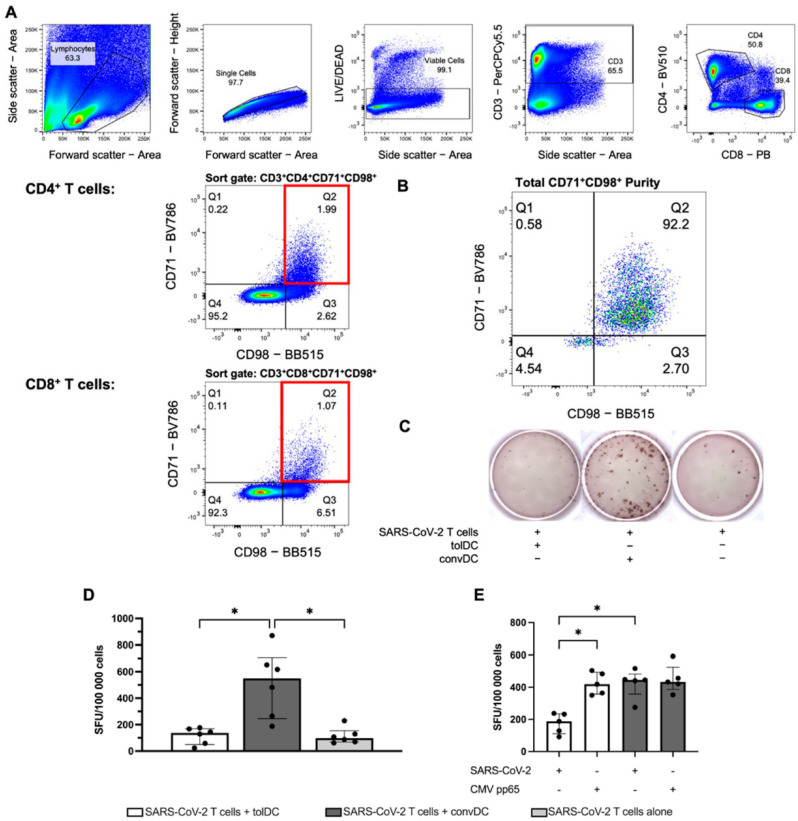
TolDC induce SARS-CoV-2-specific T cell hyporesponsiveness. (**A**) Gating strategy for the sorting of CD3^+^CD4^+^CD71^+^CD98^+^ and CD3^+^CD8^+^CD71^+^CD98^+^ cells. Cells were sorted from left to right as lymphocytes (FSC-A, SSC-A), single cells (FCS-A, FSC-H), viable cells (Fixable Near-IR Dead Cell Stain) and subsequent surface marker selection, with the sort gates indicated in red, (**B**) Representative figure of an FACS-sorted CD3^+^CD71^+^CD98^+^ T cell population indicating the purity of the sort. (**C**) Representative figure of the IFN-γ ELISpot wells from SARS-CoV-2-specific T cells stimulated with tolDC or with convDC, or SARS-CoV-2-specific T cells alone and rechallenged with SARS-CoV-2 peptides. (**D**) SFU counts normalized to SFU/100.000 cells in cocultures of SARS-CoV-2-specific T cells stimulated with tolDC (white bars) or with convDC (black bars) or SARS-CoV-2-specific T cells alone (grey bars). A *p*-value < 0.05 is indicated by * (n = 6). (**E**) Antigen-specific T cell stimulatory capacity in PBL stimulated with tolDC (white) or convDC (black) in the presence SARS-CoV-2 peptides and restimulated in an ELISpot assay with either SARS-CoV-2 peptides or CMV peptides. A *p*-value < 0.05 is indicated by * (n = 5). Abbreviations used: spot forming units (SFU), tolerogenic dendritic cells (tolDC), conventional dendritic cells (convDC), Interferon-gamma (IFN-γ).

**Figure 3 ijms-23-15201-f003:**
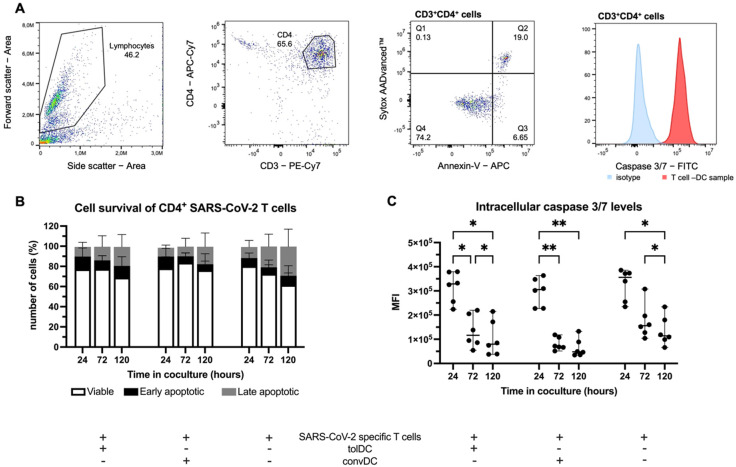
The induction of T cell hyporesponsiveness by tolDC is T cell apoptosis-independent (n = 6). (**A**) Gating Strategy for the flow cytometric follow-up of T cell apoptosis. Lymphocytes are selected based on light scatter properties (FSC/SSC), and, subsequently, CD3^+^CD4^+^ T cells are gated. Next, a quadrant gating is used to differentiate between viable cells (SytoxAADvanced^−^Annexin-V^−^), early apoptotic cells (SytoxAADvanced^−^Annexin-V^+^) and late apoptotic cells (SytoxAADvanced^+^Annexin-V^+^). The expression levels of caspase 3/7 are assessed via MFI histograms. (**B**) Follow-up of viable (white), early apoptotic (black) and late apoptotic (grey) cells in T cell–DC cocultures as compared to T cells alone by means of flow cytometry. (**C**) Mean fluorescence intensity of intracellular caspase 3/7 levels in T cell–DC cocultures at different timepoints, indicated as the median and interquartile range. Significance is indicated by * (*p* < 0.05) and ** (*p* < 0.01).

## Data Availability

The data presented in this study are available on request from the corresponding author.

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
