# Peer review of "Tolerogenic Dendritic Cells Induce Apoptosis-Independent T Cell Hyporesponsiveness of SARS-CoV-2-Specific T Cells in an Antigen-Specific Manner"

_ijms, 2022, doi:10.3390/ijms232315201_

Round 1

Reviewer 1 Report

The authors demonstrate that SARS-CoV-2 peptide-pulsed tolDC are capable of inducing SARS-CoV-2-specific T cell hyporesponsiveness in an antigen-specific manner. Furthermore, their results show that the induction of T cell hyporesponsiveness is not mediated by the induction of CD4+ T cell apoptosis. This in vitro study appears to be the first to address this topic of CD in the context of COVID-19 patients. It could be an advance in the armamentarium against this disease. Although they do not provide a complete mechanistic explanation for the induction of hyporesponsiveness of T lymphocytes by the tolDCs, the proposal is very interesting.

I have made some points throughout the PDF of the manuscript. Due to the distribution of parts of the manuscript, some doubts or questions were clarified in materials and methods. However, I have left them in the PDF to ensure that they are not omitted by the authors.

Reviewer 2 Report

1. This manuscript demonstrated the SARS-CoV-2 peptide-pulsed tolDC can induce SARS-CoV-2-specific T cell hyporesponsiveness leading to reduce the high inflammatory state of COVID-19 severe disease. The results are convinced but as for the mechanism, the authors showed that this induction is not mediated by apotosis of CD4+ T cells, while the results (Figure 3) are not conclusive. In this case, we are still wondering what is the mechanism of induction of T cell hyporesponsiveness by tolDC.

2. Please do not divide the abstract into paragraphs.
